# Check Your Shopping Cart: DNA Barcoding and Mini-Barcoding for Food Authentication

**DOI:** 10.3390/foods12122392

**Published:** 2023-06-16

**Authors:** Tommaso Gorini, Valerio Mezzasalma, Marta Deligia, Fabrizio De Mattia, Luca Campone, Massimo Labra, Jessica Frigerio

**Affiliations:** 1FEM2-Ambiente, Piazza della Scienza 2, 20126 Milano, Italy; tommaso.gorini@fem2ambiente.com (T.G.); valerio.mezzasalma@fem2ambiente.com (V.M.); fabrizio.demattia@fem2ambiente.com (F.D.M.); 2Department of Scienze Agrarie, Forestali e Alimentari, University of Turin, Via Verdi 8, 10124 Torino, Italy; marta.deligia@tiscali.it; 3Department of Biotechnology and Biosciences, University of Milano-Bicocca, Piazza della Scienza 2, 20126 Milano, Italy; luca.campone@unimib.it (L.C.); massimo.labra@unimib.it (M.L.)

**Keywords:** DNA barcoding, food fraud, species identification, food quality, food safety, food supply chain

## Abstract

The molecular approach of DNA barcoding for the characterization and traceability of food products has come into common use in many European countries. However, it is important to address and solve technical and scientific issues such as the efficiency of the barcode sequences and DNA extraction methods to be able to analyze all the products that the food sector offers. The goal of this study is to collect the most defrauded and common food products and identify better workflows for species identification. A total of 212 specimens were collected in collaboration with 38 companies belonging to 5 different fields: seafood, botanicals, agrifood, spices, and probiotics. For all the typologies of specimens, the most suitable workflow was defined, and three species-specific primer pairs for fish were also designed. Results showed that 21.2% of the analyzed products were defrauded. A total of 88.2% of specimens were correctly identified by DNA barcoding analysis. Botanicals (28.8%) have the highest number of non-conformances, followed by spices (28.5%), agrifood (23.5%), seafood (11.4%), and probiotics (7.7%). DNA barcoding and mini-barcoding are confirmed as fast and reliable methods for ensuring quality and safety in the food field.

## 1. Introduction

The complexity of the food supply network, including disruption due to COVID-19 and climate change, can make food products more vulnerable to fraud and substitution. It is difficult to quantify the impact of fraud on the whole food field because not all fraud is detected. However, food safety experts interviewed by Spielman estimated the impact of fraud on the food industry to be in excess of USD 50 billion annually [1]. Food fraud can occur anywhere in the food supply chain, from the seed supply to food packaging. Mislabelling (20.7%), artificial enhancement (17.2%), and substitution (16.4%) were the most commonly reported types of fraud [2]. Mislabelling has been frequently reported in the literature: up to 57% in processed meat products [3,4], up to 80% in fish filets [5,6], and up to 80% in dairy products [7]. Concerning the herbal supplements field, a global survey showed that 27% of herbal products commercialized in the global marketplace are adulterated. The most defrauded regions are Australia (79% mislabelled products) followed by South America (67% mislabelled products) [8]. Undeclared species substitution in food products might also represent an important health threat to allergic consumers because of the introduction of food allergens, such as different kinds of nuts and mollusks [9] or poisonous plants [10]. Even though it is a current problem, agribusiness has not paid sufficient attention to this issue. Most fraud is harmless, and this leads to a lack of attention. Nevertheless, the consumers have a large interest in the quality of food. McCallum and colleagues investigate consumers’ willingness to pay for premium products to reduce risk and uncertainty related to food fraud, showing that consumers are willing to pay for premium products to avoid food fraud and purchase an authentic product [11]. In this regard, blockchain has emerged as a promising technology that allows users to trace food products and eliminate or reduce harmful food fraud.

Treiblmaier and Garaus investigate how the use of blockchain to trace food products impacts consumers’ perception of product quality, finding that blockchain labels help to strengthen consumers’ perceived quality of food products which, in turn, increases their purchase intention [12]. Introducing DNA analysis into supply chain control could increase consumers’ confidence and consequently the budget allocated for food shopping. DNA barcoding has been frequently used in the literature for food authentication and supply chain control [13,14,15].

The application of DNA barcoding in food authentication is rooted in the concept of using short and standardized DNA sequences to differentiate between species. The technique targets specific regions of the genome, such as the mitochondrial DNA (mtDNA) or chloroplast DNA (cpDNA), which exhibit sufficient variability among species while maintaining conserved regions within the same species [16,17]. By comparing the barcode sequences obtained from unknown samples with well-curated reference databases, such as ncbi (https://www.ncbi.nlm.nih.gov/nucleotide/ accessed on 1 May 2023) and BOLD (https://www.boldsystems.org/ accessed on 1 May 2023), DNA barcoding allows for the identification of species present in food products, thereby enabling the detection of fraudulent practices.

One of the key advantages of DNA barcoding is its ability to detect adulteration and substitution in complex food matrices [18]. The technique can differentiate between closely related species or detect the presence of non-declared ingredients, even in processed or highly fragmented products. For instance, in cases where premium and expensive seafood species are substituted with cheaper alternatives, DNA barcoding can expose such fraudulent activities by identifying the true species present in the sample [19]. Similarly, it can detect the presence of allergenics that may pose health risks to consumers. Furthermore, DNA barcoding can aid in the identification of geographical origins or specific cultivars, providing valuable information regarding product quality, cultural heritage, and compliance with geographical indication regulations [20].

The use of DNA barcoding in combating food fraud has gained significant attention worldwide. Governments, regulatory agencies, and industry stakeholders recognize its potential to ensure food authenticity, protect consumer rights, and maintain market integrity. In recent years, various countries and international organizations have established initiatives and regulations to promote the adoption of DNA barcoding as a standard practice in food authentication. These include The EU Agri-Food Fraud Network (FFN), the United States Food and Drug Administration’s (FDA) GenomeTrakr program, and the International Organization for Standardization’s (ISO) guidelines on DNA-based methods for food authenticity testing.

Despite its numerous benefits, DNA barcoding is not without limitations. Challenges related to sample preparation, DNA extraction, database completeness, and the availability of suitable reference materials need to be addressed for wider adoption and successful implementation. Furthermore, ongoing advancements in DNA sequencing technologies, bioinformatics tools, and reference databases are vital to enhance the accuracy, efficiency, and reliability of DNA barcoding in food fraud detection.

This study aimed to identify several workflows of DNA barcoding for supply chain control in different food fields. A total of 38 companies, operating in 5 different fields (seafood, botanicals, agrifood, spices, and probiotics) supplied some of their high-selling products for a total of 212 specimens. Among these samples we can find fish filets, herbal teas, truffles, caviar, canned fish, processed products, powders, plant extracts, food supplements, flours, etc., a mix of products that can be considered representative of a supermarket shopping cart. The technical goals of the study are to (i) define the most suitable extraction methods for food matrices, (ii) identify the most suitable barcode region useful for different types of products (i.e., fresh, processed, etc.) and designed primer pairs when necessary, and (iii) estimate the ability of DNA barcoding tools to assess fraud in high-selling products.

## 2. Materials and Methods

### 2.1. Specimen Collection

The specimen collection was based on some defined criteria: (i) the most counterfeit species according to the literature were collected, (ii) sampling of the same species belonging to different companies was preferred, and (iii) when possible, raw/fresh, intermediate, and final products were collected. A total of 46 companies operating in the food field were contacted to join this study. The companies were chosen considering the food field operating (seafood, agrifood, spices, botanicals, and probiotic) with the aim to cover the most defrauded fields. A total of 38 companies agreed to participate in the project and a total of 212 specimens were collected (Table A1). In this study, we analyzed different typologies of products, from fresh (fish filets) to highly processed products (food supplements).

### 2.2. DNA Extraction

Considering the wide typology of specimens tested in this study, different commercial kits and extraction methods were chosen based on the literature [21,22,23,24]. For seafood specimens (fresh and intermediate specimens), Tissue Genomic DNA Extraction (Fisher Molecular Biology, Rome, Italy) (TGF) was selected and for more difficult products, such as canned fish and products preserved in oil and brine; the ReliaPrep™ gDNA Tissue MiniPrep System (Promega, Milan, Italy) (RPP) was tested/used with a modification to the protocol. The products preserved in brine were washed three times with a physiological solution (NaCl 0.7%), mixing overnight at room temperature. Canned specimens were pretreated in order to clean the tissue from the conservation liquid, such as oil (vegetable and olive); briefly, oil and lipids were removed by soaking in chloroform/methanol/water (1:2:0.8) and mixing overnight at room temperature [24]. 

For agrifood products, spices, and botanicals, a DNeasy Plant Kit (QIAGEN, Milan, Italy) (DPQ) was used following the instructions. For more complex samples belonging to these fields, such as phytoextract, the CTAB method was also applied [25]. The CTAB method allows us to start from a higher amount of material (1 g) and to harvest all the DNA in the solution.

Finally, for probiotic specimens, QIAamp DNA Microbiome Kit (QIAGEN) (QAQ) was used. In Table A1 are shown all the extraction methods used for all specimens. Purified gDNA was checked for concentration and purity by using a Qubit 2 Fluorometer and Qubit dsDNA HS Assay Kit (Invitrogen, Carlsbad, CA, USA).

### 2.3. Barcode Region Selection

A universal set of DNA barcoding markers for each product was tested. Specifically, different primer pairs were selected for animals, plants, fungi, and bacteria.

#### 2.3.1. Animal DNA Barcoding

The amplification efficiency of the barcode region is associated with the primer pairs. A primer pair specific for a universal barcode region should be versatile across a wide range of animal species and have high affinity to DNA templates. Nevertheless, sometimes the universal primers are not applicable for certain taxa or specimens and it is necessary to redesign primers, as for some specimens in this study [26]. The barcode regions chosen for animal identifications were the mitochondrial markers COI (Cytochrome c oxidase I), RNA 16S (16S ribosomal RNA), CytB (Cytochrome b), and Control Region (DLoop). For the species *Dicentrarchus labrax*, *Katsuwonus pelamis*, *Thunnus* sp., primer pairs for DNA barcoding and mini-barcoding, respectively, were designed in silico in this study. For *Dicentrarchus labrax,* the COI region was identified as the most suitable for species identification, while for *Katsuwonus pelamis* and *Thunnus* spp., the control region (CR) was chosen based on the literature [22]. All nucleotide sequences of the COI gene and control region (CR) were obtained from NCBI Nucleotide for *Dicentrarchus* spp., *Katsuwonus pelamis*, and *Thunnus* spp., respectively, and were aligned using ClustalW2 software (www.ebi.ac.uk/Tools/msa/clustalw2/ accessed on 1 May 2023). The most conserved regions for *Dicentrarchus* sp., *Katsuwonus pelamis,* and *Thunnus* spp. were identified using Bioedit software and primer pairs specific for the genus *Dicentrarchus* spp. and *Thunnus* spp. and the species *Katsuwonus pelamis* were de novo designed. Primer pairs were tested with Primer–Blast tool available from NCBI (www.ncbi.nlm.nih.gov/tools/primer-blast/ accessed on 1 May 2023) to verify the specificity. Primer sequences are shown in Table 1.

#### 2.3.2. Plant DNA Barcoding

Starting from 2005, mitochondrial, plastid, and nuclear genomes were studied to identify a barcode universal region for plants [39,40,41,42] and four gene regions (*rbcL*, *matK*, *trnH-psbA*, and ITS) have been chosen as the standard DNA barcodes in most applications for plants [43,44,45]. In this study, all of these barcode regions were tested. However, recently, some manuscripts described the efficacy of mini-barcode regions (i.e., the analysis of smaller genome portions—100–150 bp—usually associated with the largest DNA barcodes) for the identification of processed plant extracts [46,47]. Furthermore, in this study, a DNA mini-barcoding barcode (*rbcL* mini-barcoding) was tested for plant extracts. Primer sequences are shown in Table 1. The different plant regions chosen for each species were defined after an in silico analysis; the sequences for the DNA barcoding marker chosen in this study were downloaded from NCBI Nucleotide database (https://www.ncbi.nlm.nih.gov/nucleotide/ accessed on 1 May 2023). Sequences were aligned using the online tool Muscle (https://www.ebi.ac.uk/Tools/msa/muscle/ accessed on 1 May 2023) and manually edited using Bioedit. Haplotypes were collapsed by using the online tool Fabox (https://users-birc.au.dk/palle/php/fabox/ accessed on 1 May 2023). Finally, each haplotype was compared to the online database using the BLAST algorithm (https://blast.ncbi.nlm.nih.gov/Blast.cgi accessed on 1 May 2023). The best performing plant markers in terms of identification were chosen and selected for the analysis.

#### 2.3.3. Fungi DNA Barcoding

The most common barcode region for fungi identification is ITS [48,49,50]. El Karkouri and colleagues also tested this region for truffles, finding the efficiency for species identification for *Tuber* spp. Genera [51]. In this study, the ITS barcode region was also chosen for DNA barcoding analysis. Primer sequences are shown in Table 1.

#### 2.3.4. Bacteria DNA Barcoding

For bacteria identification, the 16S rRNA gene is used. It is a common housekeeping gene in all prokaryotic organisms. This gene is the most used in bacterial study because (i) it is present in almost all bacteria, (ii) the function of the 16S rRNA gene over time has not changed, suggesting that random sequence changes are a more accurate measure of the evolution, and (iii) the 16S rRNA gene (1500 bp) is large enough for informatics purposes, even if, for DNA barcoding, a smaller region is analyzed [52]. Primer sequences are shown in Table 1.

### 2.4. DNA Amplification and Identification

A standard PCR amplification was performed using PCR Mix Plus (A&A Biotechnology, Danzica, Poland) following the manufacturer’s instructions in a 25 μL reaction containing 1 μL 10 mM of each primer and 3 μL of gDNA (about 20–50 ng). PCR cycles differ in relation to the primer pairs used. All the PCR programs are shown Appendix A
Table A2. The amplicon was visualized by electrophoresis on agarose gel using 1.5% agarose Tris-acetate-EDTA (TAE) gel. Purified amplicons were bidirectionally sequenced by Sanger at Eurofins Genomics (Ebersberg, Germany). After manual editing, primer removal, and pairwise alignment, all the tested samples’ (Table A2) identities were assessed by adopting a standard comparison approach against the GenBank database with BLASTn [53]. Each barcode sequence was taxonomically assigned to the species with the nearest matches (maximum identity > 99% and query coverage of 100%).

## 3. Results and Discussion

DNA extraction was successful for 187 specimens out of 212, with high DNA quality and good yield (i.e., 3.2–27.4 ng/μL). The presences in the public databases of the sequences for all the species considered in our study were checked and confirmed. For 25 specimens (11,8% of total), 22 for botanicals and 3 for spices, the extracted DNA was not suitable for the analysis in terms of quantity and quality (Table A1). Concerning the identification, for most of the samples (88.2%), it was possible to identify the species, proving the suitability of the barcode region selected. A total of 45 samples out of 212 were defrauded for a total of 21.2% of detected fraud (Table A1).

Considering the results of this study, the most defrauded products were botanicals, with 28.8% of substitution or contamination (Table 2). To identify the contaminants, further analysis, such as Next Generation Sequencing (NGS), is necessary [54].

Almost all specimens were impossible to identify by morphological methodology, because they were treated, in the form of powder or capsule. This value is in line with the percentage presented by Ichim and colleagues, who showed that 27% of the herbal products commercialized in the global marketplace are adulterated [8]. In the same way, the higher percentage of specimens without detectable DNA (31.4%) were botanicals too (Figure 1, Table 2). This value can be explained considering that more than half of the specimens (51 of 70) undergo industrial pre-treatment, such as high temperatures, use of solvent (ethanol, glycerol), and other industrial treatments such as CO_2_ supercritical extraction. These industrial processing steps degrade, fragment, and precipitate DNA. In a previous study of ours [46], we evaluated the capability of DNA barcoding identification for botanicals (phytoextract and botanicals). We found that phytoextracts obtained through hydroalcoholic treatment, with the lower percentage of ethanol (<40%) and aqueous processing at low temperature, had a major rate of sequencing and identification success. In this study, we obtained similar results, with a success of identification for liquid aqueous phytoextracts with a low percentage of ethanol (<40%) (i.e., DIF_74, DIF_75, DIF_138, etc.) and an incapability to detect DNA in the other typology of specimens. 

After botanicals, the spice sector revealed 28.5% of defrauded products. Our results are in line with the study of Cottenet and colleagues [55]. In most of the non-compliant samples (10 of 12), we did not find a substitution, but a contamination. In some cases (DIF_147 and DIF_173), we were able to identify the genera; in all the remaining samples we obtained multiple sequences and it was impossible to identify any species or genera. This means that the contamination in those samples is high and it is possible that multiple species coexist, as indicated in the literature [56].

Although in the agrifood samples the fraud percentage is lower than botanicals and spices, we face a substitution case of fraud for all the cases. Sample DIF_193 was declared *Tuber brumale* but was identified as *Tuber melanosporum.* These two species, although similar, can be distinguished by morphological analysis. Analyzing the gleba, the *Tuber melanosporum*, known as the black truffle or Périgord truffle, it is very dark, tending to purplish-black and with fine white veins, while the *Tuber brumale*, commonly known as winter truffle or musky truffle, is grey-brownish with large and sparse veins. The interesting fact is that *Tuber melanosporum* is more expensive than *Tuber brumale.* In this case we are facing an involuntary substitution that damages the company but not the consumers. For this reason, the control of the supply chain is important, not only to offer a high-quality product, but also to avoid mistakes that can damage the company itself. The frauds detected in seafood products are not in line with the literature, which declares a percentage of 25–30% of mislabelling [57], while we detected a lower percentage (11.4%). This data can be explained considering that the specimens analyzed were collected directly from the company, assuming that all the samples were compliant. In all cases we faced a case of mislabelling, which is a false claim or distortion of the information provided on the label/packaging. The specimens DIF_009, DIF_069, and DIF_070 were different species of the same genus. They were probably an unintentional fraud. Nevertheless, the specimens DIF_008, DIF_027, DIF_028, DIF_041, and DIF_048 were found to be a totally different genus. The most serious case is the sample DIF_041. This specimen was a processed product and the species was impossible to detect by morphological analysis. It was declared as *Theragra chalcogramma* but was found to be *Lepidopsetta polyxystra. Theragra chalcogramma* belongs to the order Gadiformes and is commonly called “Alaska pollock”, while *Lepidopsetta polyxystra* belongs to the order Pleuronectiformes and is a flat fish commonly called “Northern rock sole”. The criticality of the seafood sector is that companies buy filets or semi-processed fish, unlike other sectors where the starting material is already ground or processed (e.g., botanicals, spices, etc.). This highlights a problem in the control of the supply chain. Finally, the probiotics sector was found to have the lowest percentage of fraud (7.7%). Moreover, we found that the specimen DIF_210, declared *Bifidobacterium bifidum*, was contaminated with other bacteria. There was probably an unintentional contamination in the production site with another probiotic. Recent studies have demonstrated that probiotic contamination with other probiotics is a common occurrence. For example, a study by Lewis and colleagues found that the contents of many bifidobacterial probiotic products analyzed in their study differ from the ingredient list, sometimes at a subspecies level. Only 1 of the 16 probiotics perfectly matched its bifidobacterial label claims in all samples tested [58]. The implications of probiotic contamination can vary depending on the specific strains involved and the intended use of the probiotic product. In some cases, the presence of unintended probiotics may be harmless or even beneficial. However, there is also a risk of introducing harmful or pathogenic microorganisms that may compromise the safety and efficacy of the probiotic product.

Considering the data from Table 2, it is possible to notice how processed products (such as botanicals, agrifood, and spices) have a significantly higher percentage of fraud. This is because the product, being crushed, transformed, or otherwise not in its whole form, is more difficult to identify morphologically and therefore fraud is more easily carried out.

To conclude, the extraction methods retrieved from the literature and tested in this study seem to be suitable for the chosen products, due to the DNA extraction success of 187 specimens out of 212. Moreover, most of the samples were identified at the species level, so in this study the most suitable barcode regions useful for different types of products were identified. The DNA barcoding approach, given its maturity and its wide application in the last twenty years, could be used in strategic points of the food supply chain: customs, goods management office, but also directly in medium-large companies and in the GDO. Nowadays some companies use DNA analysis to check their suppliers and to ensure customers a quality product, but this technology, although widely used in the scientific environment, is not yet fully accepted by the final consumer. Raising awareness and citizen science will be needed to convey the importance and potential of this approach. In conclusion, this study contributes to the growing body of research on DNA barcoding for species identification in the food industry.

## 4. Conclusions

Given the results of this study, DNA analysis provides a powerful tool for detecting and identifying contaminants in commercial food products, enabling manufacturers and regulatory authorities to take appropriate action to ensure the quality and safety of these products. The results confirm the suitability and reliability of DNA barcoding and mini-barcoding as fast and effective methods for ensuring quality and safety in the food field. Moreover, techniques such as LAMP, RPA, BAR-RPA, Bar-HRM, and minION have made DNA-based methods more affordable, as they require cheaper instruments and protocols [22,59,60]. However, some challenges, in particular in relation to non-conformances observed in botanicals and spices, remain an issue to investigate. Nevertheless, by addressing these technical and scientific issues and implementing standardized workflows, DNA barcoding can play a crucial role in combating food fraud and enhancing traceability in the food supply chain, thus ensuring consumer confidence and facilitating regulatory compliance.

## Figures and Tables

**Figure 1 foods-12-02392-f001:**
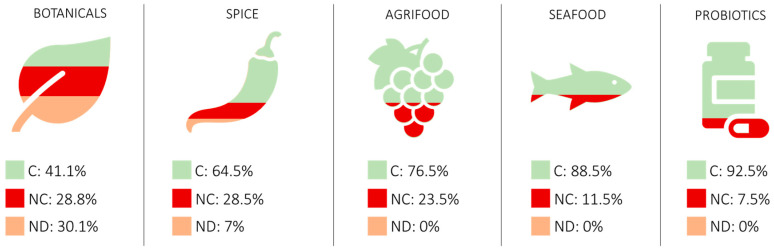
Infographics represent the percentage of compliant, non-compliant, and specimens with no DNA detected (C, NC and ND) for all the field analyzed.

**Table 1 foods-12-02392-t001:** List of primer name, gene target, primer sequence (5′->3′), bp of the fragment obtained, annealing temperature, taxonomic target, and reference.

Primer Name	Gene	Primer Sequence (5′->3′)	bp	Ta °C	Target	Reference
Cox1_Ward_FishF1	COI	F: TCAACCAACCACAAAGACATTGGCAC	655	55 °C	Bony fish	[27]
Cox1_Ward_FishR1		R: TAGACTTCTGGGTGGCCAAAGAATCA				
Cox1_Ward_FishF2	COI	F: TCGACTAATCATAAAGATATCGGCAC	616	55 °C	Bony fish	[27]
Cox1_Ward_FishR2		R: ACTTCAGGGTGACCGAAGAATCAGAA				
LCO 1490	COI	F: GGTCAACAAATCATAAAGATATTGG	700	47 °C	Crustaceans and cephalopods	[28]
HCO 2198		R: TAAACTTCAGGGTGACCAAAAAATCA				
16sar-L	16S rRNA	F: CGCCTGTTTAYCAAAAACAT	571	57 °C	Animal universal	[29]
16sbr_H		R: CCGGTCTGAACTCAGATCACGT				
GLUDG	Cytb	F: TGACTTGAARAACCAYCGTTG	1140	52 °C	Animal universal	[30]
C61221H		R: CTCCAGTCTTCGRCTTACAAG				
Tuna_CR_F	CR	F: GCAYGTACATATATGTAAYTACACC	236	58 °C	*Thunnus* spp.	[31]
Tuna_CR_R		R: CTGGATGGTAGGYTCTTACTGCG				
Tuna_CR_F	CR	F: GCAYGTACATATATGTAAYTACACC	80	52 °C	*Thunnus* spp.	[31]/This study
Tuna_minibar_R2		R: GAYATATGAATAKTTWSRTAC				
Sco5S_F	ITS	F: CTCACTGTTACAGCCTG	120	48 °C	*Scomber* spp.	[19]
Sco5S_R		R: CAAACACATGCTATCCTT				
Katw_F	CR	F: GCGAGATYTAAGACCTACCACG	80	54 °C	*Katswonus* spp.	This study
Katw_R		R: GAGCTGGTTGGTCTCTT				
Dlab_F	COI	F: TCTTATTCTCCCCGGGTTCG	186	59 °C	*Dicentrarchus* spp.	This study
Dlab_R		R: GATGTGAAGTATGCGCGTGT				
rbcL_1F	rbcL	F: ATGTCACCACAAACAGAAAC	743	50 °C	Plants universal	[31,32]
rbcL724R		R: TCGCATGTACCTGCAGTAGC				
rbcL 1	rbcL	F: TTGGCAGCATTYCGAGTAACTCC	226	50 °C	Plants universal	[33]
rbcL B		R: AACCYTCTTCAAAAAGGTC				
matK_3F_KIM	matK	F: CGTACAGTACTTTTGTGTTTACGAG	636	53 °C	Plants universal	[34]
matK_1R_KIM		R: ACCCAGTCCATCTGGAAATCTTGGTT				
psbA	psbA-trnH	F: GTTATGCATGAACGTAATGCTC	300–600	53 °C	Plants universal	[35]
trnH		R: CGCGCATGGTGGATTCACAATCC				
ITS-p5	ITS	F: CCTTATCAYTTAGAGGAAGGAG	300–750	55 °C	Plants universal	[36]
ITS-u4		R: RGTTTCTTTTCCTCCGCTTA				
ITS3_KYO2	ITS	F: GATGAAGAACGYAGYRAA	300–500	55 °C	Fungi	[37]
ITS-4		R: RGTTTCTTTTCCTCCGCTTA				
P0	16S rRNA	F: GAGAGTTTGATCCTGGCTCAG	1540	54 °C	Bacteria	[38]
P6		R: CTACGGCTACCTTGTTACGA				

**Table 2 foods-12-02392-t002:** In the table are indicated the number of specimens analyzed divided into compliant, non-compliant, and samples where DNA was not detected, also expressed in percentage.

Specimens’ Typology	Collected Specimens	Sector	Compliant (Percentage)	Non-Compliant (Percentage)	No DNA Detected (Percentage)
Fresh/raw	6	Seafood	6 (100%)	/	/
Intermediate	4		4 (100%)	/	/
Processed	60		52 (86.6%)	8 (13.3%)	/
Total	70		62 (88.5%)	8 (11.5%)	/
Fresh/raw	19	Botanicals	14 (73.7%)	5 (26.3%)	/
Intermediate	3		3 (100%)	/	/
Processed	51		13 (25.5%)	16 (31.3%)	22 (43.2%)
Total	73		30 (41.1%)	21 (28.8%)	22 (30.1%)
Fresh/raw	13	Agrifood	12 (92.3%)	1 (7.7%)	0
Intermediate	/		/	/	0
Processed	4		1 (25%)	3 (75%)	0
Total	17		14 (76.5%)	3 (23.5%)	/
Fresh/raw	18	Spice	13 (72.2%)	4 (22.2%)	1 (5.6%)
Intermediate	/		/	/	/
Processed	24		14 (58.3%)	8 (33.3%)	2 (8.3%)
Total	42		27 (64.5%)	12 (28.5%)	3 (7%)
Fresh/raw	/	Probiotics	/	/	/
Intermediate	/		/	/	/
Processed	13		12 (92.5%)	1 (7.5%)	/
Total	13		12 (92.5%)	1 (7.5%)	/

## Data Availability

The data presented in this study are available on request from the corresponding author.

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
