# Peer review of "Check Your Shopping Cart: DNA Barcoding and Mini-Barcoding for Food Authentication"

_foods, 2023, doi:10.3390/foods12122392_

Round 1

Reviewer 1 Report

Table 2, Total Botanicals is 73, not 70. So, 30 (41.1%), 21 (28.8%), 22 (30.1%).

Considering the results of this study, the most defrauded products were botanicals, with 30% (28.8%) of substitution or contamination. How can you tell if the sample is contaminated? And contaminated by what (Table A1)?

Can you provide sequences of all the samples?

Author Response

Dear reviewer,

   thank you for the comment. We have modified the percentage throughout the manuscript and changed the image. 

Regarding contamination, despite having good DNA quality, we obtained sequences with multiple double peaks. For this reason, we deduced that the samples were contaminated.

About the sequences, we have decided not to submit them to Genebank but, if necessary, we can provide them all.

Reviewer 2 Report

Dear authors,

the specific comments are included in the attached pdf file.

Author Response

Dear reviewer,

 thank you for your comments. We modified the manuscript following your advices.

Reviewer 3 Report

Scientifically, I have no problem with what the authors of this manuscript have done. They have used previously established DNA extraction techniques and PCR primers to identify fraudulent behavior in five sectors of commercial food production. My problem is simply that this has all been done before. There is no real comparison of DNA extraction methods across the five sectors, or comparison of the success of differing loci and/or sets of PCR primers for detecting fraud. In its present format, the manuscript presents nothing new.

English is reasonable but needs to be revised by a native English-speaker.

Author Response

Dear reviewer,

   thank you for your comment. In this study we tried to identify the best way to analyse processed food products. Moreover, the design of three couple of primer (for Katswonus sp, Dicentrarchus sp and Thunnus sp.) give novelty to the study. Probably it is not highlighted enough in the text, so we have modified it.